# The Role of Estrogen and Estrogen Receptors in Head and Neck Tumors

**DOI:** 10.3390/cancers16081575

**Published:** 2024-04-19

**Authors:** Jacqueline-Katrin Kranjčević, Josipa Čonkaš, Petar Ozretić

**Affiliations:** Laboratory for Hereditary Cancer, Division of Molecular Medicine, Ruđer Bošković Institute, Bijenička cesta 54, 10000 Zagreb, Croatiajconkas@irb.hr (J.Č.)

**Keywords:** head and neck tumors, HNSCC, estrogen, estrogen receptor, ERα, ERβ, membrane estrogen receptors, ERα36, GPER1, NaV1.2

## Abstract

**Simple Summary:**

Head and neck tumors (HNTs) represent a diverse group of malignancies that originate from the lining tissue of organs in the upper parts of the respiratory and digestive tracts. Given that these include parts of the body that enable breathing, speech, and feeding, these tumors cause a great reduction in the quality of life. The main risk factors are consumption of tobacco products and alcohol, as well as infection with human papillomaviruses. However, regardless of the risk factors, men have a significantly higher risk of developing HNTs than women. It is, therefore, presumed that exposure to the female sex hormone estrogen in women could provide protection against the development of those tumors in women. In this scientific literature survey, we provide a comprehensive overview of the role of estrogen and its receptors in HNTs and assess the possible benefits of therapy with either exogenous estrogen or anti-estrogens.

**Abstract:**

Head and neck squamous cell carcinoma (HNSCC) is the most common histological form of head and neck tumors (HNTs), which originate from the epithelium of the lips and oral cavity, pharynx, larynx, salivary glands, nasal cavity, and sinuses. The main risk factors include consumption of tobacco in all forms and alcohol, as well as infections with high-risk human papillomaviruses or the Epstein–Barr virus. Regardless of the etiological agent, the risk of developing different types of HNTs is from two to more than six times higher in males than in females. The reason for such disparities probably lies in a combination of both biological and psychosocial factors. Therefore, it is hypothesized that exposure to female sex hormones, primarily estrogen, provides women with protection against the formation and metastasis of HNTs. In this review, we synthesized available knowledge on the role of estrogen and estrogen receptors (ERs) in the development and progression of HNTs, with special emphasis on membrane ERs, which are much less studied. We can summarize that in addition to epidemiologic studies unequivocally pointing to the protective effect of estrogen in women, an increased expression of both nuclear ERs, ERα, and ERβ, and membrane ERs, ERα36, GPER1, and NaV1.2, was present in different types of HNSCC, for which anti-estrogens could be used as an effective therapeutic approach.

## 1. Introduction

While sex hormone receptors play a major role in certain types of tumors, such as breast [1] and prostate cancer [2], their role is also being increasingly investigated in presumably hormone-independent cancers, like head and neck tumors (HNTs). These include tumors of the upper aerodigestive tract, and the main risk factors include consumption of tobacco in all forms [3] and alcohol [4], as well as infections with human papillomaviruses [5] or the Epstein–Barr virus [6]. However, it has been observed that, regardless of the etiological agent, men in general have a two to sixtimes higher risk of developing HNTs than women [7]. Understanding how cancer affects men and women differently is a pivotal aspect of cancer research nowadays. Differences in the rates of incidence and mortality for various types of cancer highlight the importance of exploring these gender-specific disparities. The varying rates of cancer between genders are influenced by genetic and molecular mechanisms, as well as the effects of hormones such as estrogen [8]. Nonetheless, apart from the direct influence of sex hormone receptors, the emergence of HNSCC entails a complex interaction involving sex chromosomes, sex hormones, and various biological and lifestyle elements [9,10,11]. Sex hormones play a pivotal role as gene expression modulators, influencing genetic and hormonal variances that ultimately dictate the efficacy of chemotherapy [12]. Furthermore, at the genetic and molecular levels, variations in gene polymorphisms and activity of enzymes related to drug metabolism contribute to differing rates of cancer incidence observed between males and females [13]. Therefore, after surveying the literature on the role of androgen receptors [14], in this review we provide a comprehensive overview of the role of estrogen and estrogen receptors in HNTs, since exposure to estrogen in women is thought to protect against the development and metastasis of HNTs [15].

## 2. Head and Neck Tumors

HNTs represent a heterogeneous group of tumors and the most common histological form is head and neck squamous cell carcinoma (HNSCC), which accounts for more than 90% of cases [16]. The anatomical origins of HNSCC are the epithelium of the lips and oral cavity, nasopharynx, oropharynx and hypopharynx, larynx, salivary glands, nasal cavity, and sinuses [7]. Given that HNTs include parts of the body that enable breathing, speech, and feeding, this form of tumor is accompanied by a great reduction in the quality of life of the affected person [17].

HNTs are the sixth most common cancers in the world. Globally, more than 830,000 cases are diagnosed annually, of which over 400,000 die [18]. In 2020, 1003 new cases were diagnosed, and 476 deaths related to HNTs occurred in Croatia [19]. Despite medical advances in treatment, HNTs are still characterized by a significant mortality rate. HNTs account for 5.7% of all cancer-related deaths worldwide [20]. The 5-year survival rates range between 40% and 50% and, in patients with recurrent/metastatic cancer, the average survival time is only 10–13 months [21]. In addition to deaths directly caused by HNTs, there is a relatively high rate of suicide among survivors compared to other types of cancer, and the likely reasons for this are a reduced quality of life and psychological distress [22].

The common goal of the growing number of studies in the field of HNTs is to better understand their characteristic biological processes, to identify new biomarkers that would enable the prediction of the development and progression of the disease, and the discovery of new targeted therapies, which would be more effective and less toxic than the existing ones. Currently, the most common forms of treatment are surgical removal, radiotherapy, and chemotherapy, which are not always effective and can greatly affect the patients’ quality of life [23].

The International Agency for Research on Cancer (IARC), a part of the World Health Organization (WHO), has classified risk factors for the development of HNSCC, which include alcohol and tobacco consumption, and infection with human papillomavirus (HPV) or Epstein–Barr virus (EBV) [24]. 

However, the most significant risk factor for the development of cancer in general, including HNSCC, is the consumption of tobacco products in any form [25]. Tobacco contains many carcinogenic substances which lead to the formation of DNA adducts throughout the genome. The resulting damage, if not repaired, leads to mutations [26]. Consumption of tobacco products is specifically associated with mutations in the *TP53* gene, which codes for the tumor-suppressor protein p53 with a key role in maintaining genomic stability [27,28]. According to data from the International Head and Neck Cancer Epidemiology Consortium (INAHNCE), smoking up to three cigarettes a day increases the risk by 50%, while smoking three to five cigarettes increases the risk by more than double [29]. The influence of electric cigarettes on the development of HNSCC is not yet known and will only be established in the coming decades.

Alcohol consumption is another important risk factor in addition to tobacco, and it has been found that 70 to 80% of HNSCC cases are associated with alcohol or tobacco consumption [24]. Ethanol and its first metabolite acetaldehyde (associated with the consumption of alcoholic beverages) are classified as human carcinogens (Group 1) [25]. Acetaldehyde is known to form DNA adducts, which can induce mutations [7]. It has also been established that alcohol acts synergistically with tobacco in increasing the risk of HNSCC; that is, their joint effect is greater than the sum of the separate effects. Possible reasons for this synergy are the observed increased production of acetaldehyde in smokers compared to non-smokers and an increase in the permeability of the mucosa due to contact with ethanol, which increases the penetration of potential carcinogens [24]. One of the risk factors prevalent in the population of Southeast Asia and India is the chewing of betel nut products (*Areca catechu* seeds) and the leaves of the *Piper betle* plant [7,24].

Among the genetic factors that contribute to the increased risk and susceptibility to HNSCC, Fanconi anemia should be singled out. It is a rare hereditary syndrome, which occurs equally in males and females and in which mutations in FANC genes, which code for proteins of DNA repair mechanisms, lead to genomic instability. Other genetic predispositions include frequent polymorphisms in some other genes involved in the mechanism of DNA repair, cell cycle control, and apoptosis, as well as genes involved in the metabolism of aforementioned carcinogens [7,24].

Depending on the etiological agent (carcinogen or virus), HNTs can be divided into HPV-positive (HPV+) and HPV-negative (HPV−) HNSCC. The HPV virus, specifically its high oncogenic risk types, is classified as a carcinogen (Group 1) by the IARC [25]. The primary cause of HPV-positive HNSCC is the HPV-16 type, while HPV-18, HPV-31, HPV-33, and HPV-52 types are detected in a smaller percentage of patients. The carcinogenic effect of HPV originates from its oncogenes E6 and E7, whose protein products in the host cell interact with two tumor-suppressor proteins, p53 and pRb, and induce their degradation, which disrupts the cellular functions of DNA repair, survival, and proliferation [24,30]. HPV-positive HNSCC preferentially affects the oropharynx, primarily the tonsils and the base of the tongue, while tumors of the oral cavity and larynx can mostly be attributed to the consumption of tobacco, alcohol, or both. HPV-positive HNSCC shows different clinicopathological characteristics than HPV-negative and generally has a more favorable outcome [7,30]. Apart from HPV, the virus that has been shown to be a risk factor for HNTs, more precisely nasopharyngeal carcinoma, is the Epstein–Barr virus [24].

## 3. Sex-Related Disparities in HNSCC Incidence and Mortality

Regardless of the etiological agent, it has been observed that men have a two to more than six times higher risk of developing different type of HNSCC than women (Figure 1) [18]. This ratio cannot be fully attributed to differences in behavior such as tobacco smoking and alcohol consumption or the rate of HPV infection, so the reason for such disparities probably lies in a combination of both biological and psycho-social factors. Therefore, the hypothesis is that exposure to female sex hormones, primarily estrogen, provides women with protection against the formation and metastasis of HNTs [15].

## 4. Estrogens

Estrogens are the primary female sex hormones and as such are responsible for regulating the functions of the female reproductive system, as well as the development of secondary sex characteristics. The term ‘estrogens’ refers to a group of female hormones, consisting of estrone (E1), estradiol (E2), estriol (E3), and estetrol (E4). Like other steroids, their structure is built on a skeleton of 17 carbon atoms, connected in 4 fused rings: three cyclohexanes and one cyclopentane (Figure 2a). All 4 estrogens contain 18 carbon atoms and belong to the group of C18 steroids. They contain one phenolic hydroxyl group and one ketone group (E1) or one (E2), two (E3), or three (E4) hydroxyl groups on the cyclopentane ring (Figure 2b). Thanks to their similar structure, all four estrogens can bind to both nuclear and membrane estrogen receptors, but with different affinities and strength of response [31].

E2 is the predominant form of estrogen in the reproductive period of women, and for this reason the name ‘estrogen’ is often used instead of ‘estradiol’. E3 and E4 are associated with the state of pregnancy, while E1 is the primary estrogen in women after menopause [31]. Estrogen synthesis in women takes place in the ovaries and, to a lesser extent, in the adrenal glands and adipose tissue. Men also synthesize estrogen in the cells of the reproductive tract, albeit in much smaller amounts. Men and menopausal women mainly depend on the local synthesis of estrogen in extragonadal tissues [31,32]. As with other steroid hormones, estrogen synthesis begins with dietary cholesterol as the primary substrate, more precisely with low-density lipoprotein (LDL) cholesterol. The biosynthesis pathway of estrogen is shown in Figure 3 [31], while a summary of the serum and urine estradiol levels in men compared to women, at different stages of life is presented in Table 1 [33].

## 5. Estrogen Receptors

Estrogens perform their function by binding to estrogen receptors (ERs), which can be divided into two groups according to their subcellular localization: nuclear and membrane [34,35]. Binding of estrogen leads to the activation of the transcription of certain genes and/or signaling cascades that ultimately affect gene expression. Thus, the mode of action of the estrogen receptor can be divided into genomic and non-genomic effects [31], which will be further described.

### 5.1. Nuclear Estrogen Receptors

Nuclear estrogen receptors (nERs), ERα and ERβ, are members of the superfamily of nuclear receptors, which are also transcription factors. This family includes steroid receptors, receptors for thyroid hormones, retinoids, vitamin D, liver receptor X, peroxisome proliferator-activated receptors, and orphan receptors, for which no ligand has yet been identified [36]. Steroid nuclear receptors share significant structural homology (highly conserved DNA-binding domains and less conserved ligand-binding domains) (Figure 4), which points to their evolutionary connections and explains the similarities in the mechanisms of DNA binding and transcriptional regulation among members of the nuclear receptor superfamily [31,35].

The human ERα is encoded by the *ESR1* gene, which is located on chromosome 6 [37]. It occurs in multiple isoforms: the full-length 66 kDa isoform and several shorter ones, which are the product of alternative splicing, or the presence of alternative START codons. The truncated isoforms lack either the AF-1 or both the AF-1 and AF-2 regions, which results in their limited transcriptional activity [31]. They are, however, able to dimerize with the full-length isoform, inhibiting its AF-1-mediated activity [38,39].

ERβ is encoded by the *ESR2* gene, which is located on chromosome 14 [40]. It also occurs in multiple isoforms: the full-length 59 kDa isoform, several shorter ones, but also the elongated 61 kDa isoform. The shorter isoforms differ from the full-length protein mainly in their C-terminal LBD, affecting their ligand-binding and transcriptional ability. In addition, it has been shown that they preferentially heterodimerize with the full-length ERα, rather than ERβ, thereby repressing its transcriptional activity [41]. This is one example of the complex interplay and diverse influence that two nERs have on estrogen signaling.

### 5.2. Membrane Estrogen Receptors

In addition to the well-studied classical nuclear estrogen receptors, emerging research has illuminated the existence and significance of putative membrane estrogen receptors (mERs), whose activation leads to rapid cellular responses (Figure 5). Several studies have shown that some part of the classical ERα and ERβ, as well as their splice variants, can be found at the plasma membrane, after having been through post-translational modifications like S-palmitoylation [42] or myristoylation [39]. Other proposed mERs include the G protein-coupled estrogen receptor 1 (GPER1) [43], the voltage-gated sodium channel NaV1.2 [44], Gq-mER [45], and ER-X [46]. In this review, a closer look will be taken only at mERs that have been associated with HNTs.

#### 5.2.1. ERα36

ERα36 is a 36 kDa truncated variant of the full-length ERα66, encoded by the same *ESR1* gene. Identified in 2005 [39], the shortest known isoform is a result of the presence of an alternative promoter that resides in the first intron, which indicates that the regulation of expression of ERα36 is distinct from ERα66. This is in line with the observed expression of ERα36 in ER-negative MDA-MB-231 and MDA-MB-436 breast cancer cell lines that lack ERα66 expression [47]. ERα36 lacks both transactivation domains AF-1 and AF-2 and, therefore, possesses no intrinsic transcriptional activity. It does, however, contain the intact DNA-binding domain, as well as the partial dimerization and ligand binding domains (LBD). Its LBD has a unique 27 amino acids domain, which may broaden its ligand binding spectrum in comparison to ERα66. The ability to bind to the same DNA sequences as ERα66 and ERβ, but lack of intrinsic transcriptional activity, explains the inhibitory effect ERα36 has on both estrogen-dependent and estrogen-independent transactivation activities of ERα66 and ERβ, and makes ERα36 a negative regulator of genomic estrogen signaling mediated by the classical nERs [47]. ERα36 predominantly resides at the plasma membrane, possibly due to three potential myristoylation sites located near the N-terminus, where it mediates rapid membrane-initiated estrogen signaling [47,48,49,50,51].

#### 5.2.2. GPER1

The G protein-coupled estrogen receptor 1 (GPER1) is a typical G protein-coupled receptor (GPCR) and as such consists of seven transmembrane α-helices, four intracellular, and four extracellular loops. The extracellular loops are responsible for the recognition and binding of ligands, while the intracellular segments bind G-proteins, the activation of which leads to the initiation of various intracellular signaling pathways [31,52]. In contrast to the usual GPCRs, a significantly smaller proportion of GPER1 is localized on the cell membrane, and the primary place of its localization are the membranes of the endoplasmic reticulum and the Golgi apparatus [53]. However, its localization varies dynamically, depending on specific environmental signals, and the distribution is specific to the type of cells and tissues [54]. GPER1 is encoded by the *GPER1* gene, previously named *GPR30*, which is located on chromosome 7 [55]. After it was established that the treatment of cells with estrogen leads to the activation of the MAPK signaling pathway via GPR30 in the year 2000 [43], and five years later after it was proven that estrogen specifically binds to GPR30 [56,57], the orphan receptor GPR30 was given the new name GPER1, which recognizes its role as an mER.

Since it is a membrane protein with a dynamic structure, its purification and crystallization are difficult, and its complete crystallographic 3D structure is not yet available. However, by means of homology modeling using the GPCR-I-TASSER server for predicting GPCR protein structures, a high-quality model of GPER1 was obtained [58]. Using molecular docking simulations, a binding pocket in the exoplasmic and/or transmembrane region of GPER1 was predicted as a binding site for E2.

#### 5.2.3. NaV1.2 (SCN2A)

NaV1.2 (encoded by the *SCN2A* gene, located on chromosome 2) is a voltage-gated sodium (NaV) ion channel [59]. It is a membrane glycoprotein complex, which changes its conformation in response to an initial membrane depolarization and opens a transmembrane pore, through which Na^+^ ions selectively pass down their electrochemical gradient. The resulting Na^+^ current initiates an action potential in excitable cells like neurons and muscle cells. However, non-excitable cells can also express NaV channels, where they assume noncanonical roles that are not related to the generation of action potentials. Eukaryotic NaV channels are built from one large pseudo-tetrameric α-subunit (260 kDa), which forms a transmembrane pore, associated with one or two auxiliary β-subunits (30–40 kDa). The α-subunit consists of one large polypeptide chain, which forms four homologous domains (DI-DIV), each of which contains six transmembrane α-helices (S1–S6) and a voltage sensor. In humans, there are nine different α-subunits (NaV1.1–NaV1.9), encoded by the genes *SCN1A*, *SCN2A*, *SCN3A*, *SCN4A*, *SCN5A*, *SCN8A*, *SCN9A*, *SCN10A,* and *SCN11A*, respectively [54,60,61].

Many drugs, as well as toxins, can bind to different sites on NaV and thus inhibit the Na^+^ flux through it [60]. Tamoxifen (TAM), which will be discussed later, is one of the ligands of nER and their selective modulator, and as such, is used in the treatment of estrogen-sensitive breast cancers. Two decades after the inhibitory effect of TAM on sodium ion channels was observed [62], Sula et al. identified the binding site of TAM and its metabolites on NaV using X-ray crystallography [44]. Two binding sites were defined near the channel opening towards the cell interior, which do not overlap with the already known binding sites of other NaV ligands. The TAM binding site in NaV was observed to show geometric similarity to that in nERs, including the side chains of one glutamate and one aspartic acid, which form key hydrogen bonds with TAM at opposite ends of the molecule. Using in vitro electrophysiological tests, they also found that TAM, as well as its metabolites, can bind with high affinity to NaV and thus inhibit the Na^+^ flux through the ion pore [44]. For this reason, NaV1.2 has been considered as an mER [34].

### 5.3. Mechanisms of ERs Action

#### 5.3.1. Genomic Effects

It is believed that the activation of gene expression in response to the binding of estrogen (or other agonists) can be achieved in two ways: (1) by direct binding of the E2-ER complex to specific DNA sequences and interaction with coactivators and components of the RNA-polymerase II initiation complex; and (2) indirectly, through the interaction of ER with other transcription factors, thus stabilizing their binding to DNA and/or recruiting coactivators to the complex, whereby ERs themselves do not bind to DNA [31,35].

As type I nuclear receptors, ERα and ERβ are located in the cytoplasm. The binding of estrogen induces a conformational change, prompting translocation of the E2-ER complex into the nucleus, where dimerization occurs. The resulting homodimer or α/β heterodimer binds to DNA sequences known as estrogen response elements (EREs) within the promoters of estrogen-responsive genes (ERGs), acting as a transcription factor [31,63,64]. The consensus ERE is a 15 bp inverted repeat: 5’-AGGTCAnnnTGACCT-3’, where ‘nnn’ denotes a three-nucleotide spacer, and ‘n’ any nucleotide. It has been shown that the variability in the ERE sequence affects the conformation or structure of the cofactor binding site on the nER, which consequently affects the preference in the recruitment of certain coactivators or corepressors. This showed that the ERE sequence itself affects the allosteric regulation of nER activity and partially explains the differences in gene expression under nER regulation [65].

Despite the similarity of the amino acid sequences of the DNA-binding domain (96% identity) and the recognition of the same canonical DNA sequence (ERE), ERα and ERβ differ in the strength of their transcriptional response. It has been shown that α/β heterodimers are weaker compared to ERα homodimers in their ability to induce transcription of ERGs. This is likely due to the significant difference in their N-terminal domain, specifically their AF-1 region, which hampers ERβ’s ability to recruit certain coactivators. For this reason, ERα and ERβ show different expression profiles and, consequently, significantly different biological effects [66,67].

However, not all ERGs contain EREs or similar sequences in their promoters—an estimated 35% of them do not. In such genes, ligand-activated ERs regulate expression by protein–protein interactions with other transcription factors at their respective response elements, without binding to DNA themselves. This transcriptional crosstalk significantly extends the regulatory influence of estrogen [31,32].

Many ERGs participate in biological processes such as cell cycle regulation, proliferation, apoptosis, cell communication, and cell adhesion [68]. In a large number of HNTs, the level of expression of nER is elevated, which indicates their possible role in the development of that type of tumor. In agreement with the previously described structural difference of the N-terminal domain of nERs, it was observed that differential expression patterns of ERα and ERβ in the same type of tumor can have opposing effects on proliferation, where ERα promotes growth and ERβ has an inhibitory effect [69].

#### 5.3.2. Non-Genomic Effects

In addition to changes at the level of transcription of target genes and protein synthesis induced by estrogen, a process that lasts for hours or even days, it has been shown that estrogen can induce certain cellular responses in a much shorter time. These findings led to the hypothesis that estrogen can act through other, non-genomic mechanisms, and the subsequent discovery of the mER GPER1 [31,32,70].

Activation of GPER1 by estrogen binding leads to the activation of many intracellular signaling pathways, most of which are preceded by transactivation of the epidermal growth factor receptor (EGFR). GPER1 promotes Ca^2+^ mobilization, which can induce tumorigenesis and metastasis. Furthermore, it activates signaling pathways of various kinases, such as the MAPK/ERK signaling pathway, which promotes proliferation; the PI3K/AKT signaling pathway, which inhibits apoptosis; or c-Jun N-terminal kinase (JNK), which in turn induces apoptosis. Like other GPCRs, GPER1 activates adenylyl cyclase, and the resulting cAMP activates protein kinase A (PKA) [53,69,71]. GPER1 can also indirectly regulate gene expression, through signaling pathways that cause phosphorylation and activation of transcription factors [53]. However, the influence of GPER1 on gene expression is much smaller than of nERs, which regulate the expression of a much larger number of genes [71].

ERα36 has also been found to transduce estrogen-dependent activation of the MAPK/ERK and PI3K/AKT signaling pathways, as well as the PKCδ/ERK signaling pathway, which are of importance for malignant properties of cancer, such as cell proliferation, metastatic potential, and protection against apoptosis [47,48,49,50,51].

It has been shown that sodium ion channels promote tumor metastasis, but the exact mechanism has not been fully elucidated. One of the possible ways is that the overexpression of NaV in tumors and the consequent influx of Na^+^ increases the activity of the Na^+^/H^+^ exchanger NHE1 (sodium-hydrogen exchanger 1). The result is a lower extracellular pH, which enhances the hydrolytic activity of cathepsin and leads to the breakdown of the extracellular matrix and promotes tumor cell invasiveness and metastasis [72].

But not only mERs are responsible for estrogen-induced non-genomic effects. It has been shown that certain variants of ERα and ERβ can interact with scaffold proteins, G-proteins, various membrane receptors, and signaling molecules. In this way, both ERα and ERβ can also activate intracellular signaling pathways that influence transcriptional regulation [31].

#### 5.3.3. Other Mechanisms of Action

Regulation of gene expression, apart from the described genomic and non-genomic mechanisms separately, can be achieved by their crosstalk, i.e., convergent signaling pathways, which include protein–protein interactions of components of both mechanistic pathways. ER activation is also possible in the absence of ligands (estrogens or other agonists). It is mainly performed by phosphorylation of the ER itself or by its association with co-regulators, which can increase or decrease the activity of the ER [31,32].

## 6. Role of Estrogen and Estrogen Receptors in Head and Neck Tumors

### 6.1. Role of Estrogen

Several studies so far have shown a protective effect of estrogen exposure for developing HNTs. Although women have a 5% lower risk of death than men for all cancers combined, even a 12% increased survival rate is reported for the HNT in a European population [73]. However, higher survival rates slightly decreased with age, suggesting that female hormones play an important role in improvement of survival, while menopause leads to the cessation of the protective effect of hormones. Peltonen et al. observed an inhibitory effect of E2 on ER-positive HSC-3 and SCC-25 tongue carcinoma cell lines, while dihydrotestosterone treatment had no effect, which also supports the hypothesis of a protective role of estrogen in women [74]. However, Robbins et al., in their study on ER-positive UM-SCC-5 and UM-SCC-11B laryngeal carcinoma cell lines, did not observe any effect of E2 on cell growth, while the opposite effect was observed in vivo, where estrogen had a stimulatory effect on tumor growth [75]. This opposite effect and contradictory results can originate due to the endogenous estrogen levels fluctuation and abnormal estrogen metabolism during HNT progression compared to healthy controls, as well as different ER isoform expressions during different stages of the disease [76]. Furthermore, as was shown in a study published by Hashim et al., higher estrogen levels were associated with a lower risk of HNT development in women who are pregnant, have given birth below 35 years of age, or have undergone hormone replacement therapy [15]. Similarly, in Freedman et al. cohort study which included 297 cases of HNTs, older age at menopause was inversely associated with HNT. On the other hand, the same study showed that the use of menopausal hormone therapy (MHT) was significantly associated with a lower risk of HNT development, but also the use of estrogen-plus-progestin MHT conferred 0.47 times the risk for HNT [77]. Another study also showed no significant association between alcohol consumption, smoking, and age at menarche or menopause with oral cavity cancer in postmenopausal women, but the oral estrogen, as well as combined estrogen plus progestin were associated with an elevated risk of oral cavity cancer, suggesting that MHT increases the risk of oral cavity cancer in postmenopausal women [78]. Interestingly, since the overexpression of ERα in HPV+ HNTs was discovered, the effect of estrogen in the same cancer type was also examined. Bristol et al. have demonstrated two potential mechanisms of action of E2 treatment on HPV+ cell growth attenuation. They have shown that either the repression of the viral transcriptional long coding region (LCR) after E2 treatment, or expression of E6 and E7 HPV genes sensitizes cells to estrogen and leads to tumor growth suppression. Therefore, E2 represents a potential therapeutical treatment for HPV+ oral cancers [79]. Interestingly, Shatalova et al. have demonstrated that exposure to estrogen inhibited the apoptosis in the premalignant MSK-Leuk1 oral leukoplakia cell line, suggesting that estrogens may be involved in the progression of premalignant lesions to HNSCC [80].

### 6.2. Role of Nuclear Estrogen Receptors

So far, an elevated expression of nERs has been observed in a large number of HNT studies. In one study on HNSCC tissue samples from different sites, an increased ERβ expression was observed in tumor tissue compared to normal tissue. There was no difference in expression between males and females, neither in tumors nor in normal tissue [80]. Interestingly, in most studies of oral squamous cell carcinoma (OSCC), ERβ was described as a predominantly expressed sub-type of nERs [81,82,83,84], although there are contradictory results in other studies where ERα expression was predominant over ERβ in both oral cavity and laryngeal/hypopharyngeal cancers [85]. In the same study, relatively frequent co-expression (in 40.3% cases) of ER and progesterone receptors independent of the primary tumor site was also shown. In another study, a higher level of both *ESR1* and ERα was observed in laryngeal cancer samples [86]. It was also shown that increased expression of each of nER has a mutually opposite effect in the same type of HNT—while ERα contributes to the growth of papillary thyroid cancer [87], ERβ shows an inhibitory effect [88]. In addition, ERβ expression in oropharyngeal carcinomas (OPSCC) is associated with a higher survival rate compared to ERβ-negative ones [89]. Similarly, it was shown that higher ERα expression is associated with improved survival rates (overall, disease-specific, progression-free, and relapse-free survival) in OPSCC patients receiving primary chemoradiation [90], and it is a biomarker for better overall survival in patients with HPV+ OPSCC [91]. On the other hand, Doll et al. in their comprehensive study demonstrated a significant influence of ERα expression on a decrease in overall and relapse-free survival only in the male OSCC cohort, in comparison to ERα-negative patients [92]. Furthermore, in another study, both HPV positivity and smaller HNSCC tumor size (≤T2) were independently associated with ERα positivity [93]. Although the exact mechanism of the decrease in survival rates of ERα-positive patients is unknown, one of the possible hormonal sources in HNSCC might be an estrogen production of inflammatory cells as a response to carcinogens, like it was previously described in lung cancer [94]. Expression and activity of focal adhesion kinase (FAK) also play an important role in the ERα phosphorylation, where its elevated expression is associated with increased ERα phosphorylation, transcription, and cell growth of OSCC cell lines. Furthermore, FAK-promoted ERα phosphorylation was eliminated by the protein kinase B (AKT) inhibition, suggesting that OSCC has functional ERα and its activity can be regulated through FAK/AKT signaling, which therefore represents a novel target for OSCC treatment [95].

Interestingly, ERβ also directly controls *NOTCH1* gene expression during differentiation through RNA polymerase II pause release, while mutations in *NOTCH1* are associated with the development of squamous cell carcinoma (SCC). In addition, both increased ERβ expression and ERβ agonists treatment were associated with the inhibition of SCC cells proliferation and promoted NOTCH1 expression in vitro and mouse xenotransplants [96]. Furthermore, ERβ was expressed in the majority of laryngeal carcinomas (83%) and its expression is in a positive correlation with the maintenance of E-cadherin and ß-catenin at cell junctions of the tumor cells plasma membrane and a negative correlation with the increased TNM stage, nuclear translocation of β-catenin, and loss of the E-cadherin [97]. Therefore, it is indicated that ERβ can protect laryngeal cancer cells from the acquisition of aggressive epithelial–mesenchymal transition (EMT) characteristics. Likewise, except for the higher expression of the ERβ, the expression level of CYP1B1, a key enzyme for estrogen metabolism, was also elevated in HNSCC tissue compared with normal epithelium, but not after the in vitro E2 treatment [80]. To sum up, different cases of the same tumor type can have significant variability in the expression of a particular form of nER, and thus a different response to the presence of E2, which contributes to the heterogeneous nature of HNTs.

### 6.3. Role of Membrane Estrogen Receptors

The role of mERs in tumors has generally not been investigated as much as the role of nERs. However, some studies point to the potential importance of mERs in HNTs. One in vitro study showed that GPER1 is responsible for the upregulation of interleukin-6 (IL-6) and is thereby involved in promoting the proliferation and migration of laryngeal squamous cell carcinoma (LSCC) cells in response to bisphenol A, an estrogen mimetic. Furthermore, specific inhibition of GPER1, but not ERα/β, reduced the observed effect, which confirms that the effect is unrelated to the action of nERs [98]. In a comprehensive bioinformatic analysis of the role of *GPER1* in cancer, an elevated expression of *GPER1* was observed in HNSCC, compared to the normal tissue, suggesting its diagnostic potential. In addition, lower expression of *GPER1* was also associated with poor prognosis, so *GPER1* may also be a prognostic marker for this cancer type [99]. Interestingly, GPER1 antagonist G15 has shown an antitumor effect in SCC-4, SCC-9, and HSC-3 human OSCC cell lines. It induces dose-dependent cytotoxicity, G2/M cycle arrest, and apoptosis, as well as downregulates the expression of AKT, cell cycle-related proteins, and mitogen-activated protein kinases. Additionally, G15 induced the formation of autophagosomes, suggesting it possesses anti-proliferative effects and, therefore, it represents a potential new approach to the treatment of OSCC [100]. 

The expression of the *SCN2A* gene in HNTs has so far been poorly investigated, but increased expression of this gene has been observed in highly metastatic ovarian tumor cells compared to low metastatic cells, which suggests its role in the regulation of migration and invasion of tumor cells [101]. On the other hand, HPV viral integration into the *SCN2A* genomic region was observed in oral and oropharyngeal cancers. Integration of the viral genome leads to the fusion of the HPV L2 gene into *SCN2A* intron 16, resulting in gene disruption and homozygous loss of the *SCN2A* locus [102]. Furthermore, an altered expression level of *SCN2A* was observed in smoking HNSCC patients compared to never-smoking patients. *SCN2A* was one of the upregulated differentially expressed genes that discriminated between these two cohorts [103].

The involvement of ERα36 in HNTs has not been explored extensively either. However, Schwartz et al. showed, in an in vitro assay using laryngeal cancer cell lines, that ERα36 increases protein kinase C (PKC) activity, which leads to increased proliferation and survival, and also enhances the expression of metastatic and angiogenic factors in response to E2. Both of these effects were blocked by ERα36 antibodies. In the same study, they also reported a positive correlation between the amount of ERα36 and vascular endothelial growth factor (VEGF) in laryngeal tumor samples, and also between ERα36 and metastasis to regional lymph nodes, suggesting that ERα36 plays a role in lymph node metastasis [50]. Recent studies by Verma et al. on LSCC have found an inverse correlation between ERα66 and ERα36 expression and clinical cancer stage, where ERα66 was decreasing with ascending tumor aggressiveness, while ERα36 expression was increasing. The overall high expression of ERα36 in LSCC samples and surrounding epithelia indicates an important role of that variant in the tumorigenesis and tumor progression of laryngeal cancer [104,105]. Table 2 summarizes the current knowledge about the role of nuclear and currently known membrane ERs in HNTs.

## 7. Antiestrogens and Phytoestrogens as a Therapy for Head and Neck Tumors

### 7.1. Impact of Antiestrogen Treatments on Head and Neck Tumors

Since estrogen has a proliferative effect on cells, hormone-dependent cancers that express ERs can be treated with hormone therapy in two ways: by inhibiting the synthesis of estrogen in the body or by blocking the effect of estrogen on cancer cells. For this purpose, various drugs are being used that can be classified into three categories: (1) aromatase inhibitors, which inhibit an enzyme that catalyzes the reaction of estrogen synthesis from androgenic substrates and in turn reduce estrogen levels; (2) selective estrogen receptor modulators (SERMs), which can act as either an agonist or antagonist of the ER, depending on the type of tissue, activating or blocking the ER, respectively; and (3) selective estrogen receptor degraders (SERDs), which, like SERMs, occupy the binding sites of estrogen in their receptors, but thereby induce their degradation and are exclusively antagonists of ER [106]. Since a large proportion of HNTs show enhanced expression of ERs, it can be assumed that antiestrogens could also have a therapeutic effect on those types of tumors.

Tamoxifen is a derivative of triphenylethylene, which belongs to the SERMs and acts as a non-steroidal antagonist and partial agonist of nERs. It has been used for decades as a therapy for hormone-sensitive breast cancers. It acts as a competitive inhibitor of the nER and thus prevents the binding of estrogen and its effect on cells that express nERs. Binding of TAM to the ligand binding site within the nER leads to similar molecular events as estrogen binding (chaperone dissociation, receptor dimerization, translocation into the nucleus, and binding to the ERE), but the resulting complex of TAM, nER, and DNA is not transcriptionally active. This is because the conformational changes are specific to each ligand, affecting the protein–protein interactions of the transcription complex. Thus, the complex of nER and TAM contains a no longer functional AF-2, which is required in addition to AF-1 for full nER activity. Partial agonistic action is achieved by AF-1, whose activity is regulated by growth factors acting through the MAPK signaling pathway and depends on the cell type and promoter context [107]. Therefore, as was noted before, since nER has two subtypes, ERα and ERβ, it was demonstrated that TAM is a pure antagonist only of ERβ, while it is a partial agonist of ERα [108]. However, Robbins et al. observed that TAM inhibits growth of laryngeal cancer cell lines UM-SCC-5 and UM-SCC-11B in vitro, at concentrations as low as 2 µM, as well as in vivo, though in that case the inhibitory effect of TAM was less pronounced [75]. Hoffmann et al. also observed an inhibitory effect of TAM on UM-SCC-11B, UM-SCC-14C, and UM-SCC-22B HNT cell lines, although all the studied cells were ER-negative [109]. To achieve the growth inhibition effect, as well as cytotoxic effects of TAM in ER-negative HNSCC cell lines, higher doses of TAM are needed compared with the dosages sufficient for the same effect in ER-positive MCF-7 breast carcinoma cell lines [109]. Grenman et al. observed insensitivity to TAM treatment in three ER-negative HNT cell lines (UM-SCC-5, UM-SCC-9, and UM-SCC-12), while it had an inhibitory effect on three ER-positive HNT cell lines (UM-SCC-1, UM-SCC-3, and UM-SCC-14B) [110]. Furthermore, a similar effect of human SCC-4, SCC-9, and SCC-25 lines proliferation inhibition by inhibiting G1/S phase progression was also described. In addition, this inhibition correlated with the p27 upregulation as well as the downregulation of cyclin E and CDK6 protein levels [111]. Except for its effect on proliferation, TAM, but not E2, also inhibited invasion of the SCCTF, SCCKN, SAS, and Ca9-22 OSCC cells and induced the anoikis in a concentration- and time-dependent manner. Anoikis was a direct result of adhesion inhibition and disruption of survival signals, due to the reduction in the phosphorylation of FAK, ERK, and MAPK [82]. Therefore, combination therapy of TAM together with chemotherapy was also observed as a potential treatment for OSCC cell lines. Namely, a combination treatment with TAM and cisplatin enhanced the cytotoxicity and apoptotic effect on A-253, HSC-3, and KB OSCC cells, possibly through the inhibition of PKC activity [108]. The authors also emphasized the possible mechanism of cytotoxic and growth-inhibitory effects of TAM treatment in OSCC cell lines is the upregulation of transforming growth factor beta-1 proprotein (TGFB1) since its level increased 24 h after the TAM treatment. Almost the same effect was observed in HN-6 and HN-5 OSCC cell lines, in which TAM also induced a G1 cell cycle arrest independently of p53 status and resulted in an increased level of hypophosphorylated active RB. Furthermore, combined with cisplatin, TAM also induced apoptosis more effectively and resulted in increased secretion of TGFB1 [112]. In one study, even the delay of cisplatin resistance development was demonstrated in the presence of TAM, unrelated to the ERs expression, the number of antiestrogen binding sites, or the affinity of TAM for these binding sites but linked only to the nature of the interaction between these two compounds [113]. Interestingly, Nelson et al. demonstrated the growth inhibition and induction of the OSCC cell aggregation ability after TAM treatment was not in association with the changes in E-cadherin or β-catenin expression [114]. As for mERs, it has been shown that TAM binds to GPER1 with high affinity and acts as its agonist, activating the G-protein and thereby mimicking the effect of estrogen on GPER1. As mentioned previously, TAM also binds with high affinity to NaV1.2, which results in the inhibition of the flow of sodium ions through the channel. TAM also acts as an agonist of ERα36, activating the MAPK/ERK, PI3K/AKT, and Src/EGFR/STAT5 signaling pathways and stimulating cell growth [47,48,115]. Accumulating evidence suggests that the agonistic effect of TAM on membrane-initiated signaling pathways is an important cause of tamoxifen resistance [116,117].

Fulvestrant (FULV) is an nER antagonist and, compared to TAM, has no agonistic effects described. It was first approved in the USA as the treatment of postmenopausal, hormone receptor-positive women with progressive metastatic breast cancer after antiestrogen therapy [118]. After binding to the nER, it prevents receptor dimerization and blocks the nuclear localization of the receptor [119]. Furthermore, in the FULV-ER complex, both AF-1 and AF-2 regions are inactive, which makes this complex transcriptionally inactive. Therefore, FULV acts as an SERD by binding, blocking, and accelerating the ER degradation, which ultimately results in complete inhibition of genomic estrogen signaling [118]. However, this is not the case for ERα36, where FULV failed to induce its degradation, and the reason for this possibly lies in the truncated LBD of ERα36 [39,120]. In FaDu cell line both FULV and TAM significantly sensitized tumor cells to fractionated irradiation (IR). Therefore, a possible combination therapy of TAM or FULV with radiotherapy for HNSCC patients is observed, since the same study showed that HNSCC cells with combined expression of *ESR2* and gene for submaxillary gland androgen-regulated protein 3A (*SMR3A*) have a higher risk for radiotherapy failure [121]. Similarly, treatment with both TAM and FULV inhibited the growth of UM-SCC-14A, UM-SCC-14B and UM-SCC-14C OSCC cell lines, as well as reduced β1 integrin transcription and α3 integrin cell surface expression (TAM) and laminin-1 adhesion (FULV) [122]. As the ability to decrease apoptosis after the E2 treatment was described in the premalignant HNSCC cell lines, the antagonizing effect of FULV was also described in the same cell lines. Namely, treatment with FULV can restore the estrogen-dependent apoptosis in MSK-Leuk1 premalignant cell lines, suggesting the beneficial role of antiestrogens in the treatment of HNSCC [80].

Centchroman is another SERM first synthesized at the Central Drug Research Institute, India, as a nonsteroidal oral contraceptive [123]. Apart from its role as a contraceptive pill, it acts as a competitive antagonist of the ER and promotes the conversion of E2 to a less active form, E1 [124]. Although its role was investigated primarily in target tissues, e.g., endometrium, in a study by Srivastava et al., an antiproliferative effect of centchroman in FaDu, CAL-27, SCC-25, and SCC-9 HNSCC cell lines was described. In addition, they have shown that centchroman inhibits cell proliferation in a dose-dependent manner, as well as induces apoptosis and inhibits AKT/mTOR and STAT3 signaling. Furthermore, it inhibited the colony formation of HNSCC cells and altered proteins associated with DNA damage and cell cycle regulation [125]. Therefore, centchroman also acts as a promising therapeutic candidate for HNSCC treatment. Table 3 summarizes the current knowledge about antiestrogens as potential therapies for head and neck tumors.

### 7.2. Phytoestrogens in Head and Neck Tumor Prevention, Treatment and Pathogenesis

Apart from the well-known commercial or synthetic antiestrogens, many compounds extracted from plants, so called phytoestrogens, have also shown the inhibition of estrogenic effects [126]. Therefore, phytoestrogens can also be involved in the prevention and treatment of hormone-dependent cancers by acting directly or indirectly on ERs. Compared to synthetic antiestrogens, phytoestrogens have many nutritional and pharmacological advantages and may have positive effects on cell cycle regulation and signaling, inflammation, as well as an oxidative stress response [127]. Furthermore, the biggest advantages of all natural phytochemicals are their low side effects and high safety profile, though their low bioavailability limits their usage in clinical practice, indicating the need for different drug delivery methods to improve their bioavailability [128]. However, since both natural and synthetic phytoestrogens have some limitations in disease treatment, a combined therapy using both could result in a higher efficacy of synthetic drugs. Nowadays, numerous phytochemicals such as flavones, isoflavones, flavonoids, polyphenols, phenolic acids, lignans, tannins, stilbenes, coumarins, sterols, and terpenoids have been pointed out as pure ERs agonists, as partial agonists, or as pure antagonists [129].

One of the most investigated phytoestrogen compounds in the context of HNTs is the isoflavone genistein. Genistein was initially extracted from the *Genista tridentata* L. plant; however, it is also present in a variety of soy-based food products [130]. Amongst its numerous benefits for human health, genistein proved to be effective in time-dependent and irreversible proliferation inhibition of the HNSCC cell line HN4, as well as in cell cycle arrest, by arresting the cells at S/G2-M phases and inducing apoptosis in the same cell line [131]. In their further investigation, the same group of authors has shown that the above changes were accompanied and supported by cyclin-dependent kinase 1 (CDK1) and cyclin B1 down-regulation, as well as up-regulation of the CDK inhibitor p21WAF1 and apoptosis regulatorBAX, which altogether supports their previous findings [132]. Likewise, Dev et al. have demonstrated that treatment with genistein nanoparticles selectively induced apoptosis in the OSCC cell line JHU-011 compared to the normal fibroblast cells, and this effect was accomplished by increasing the production of reactive oxygen species (ROS), leading to the translocation of BAX proteins to the mitochondria and the activation of caspase 3 [133]. The effect of genistein treatment on the cell cycle and apoptosis was also observed in several papers by Alhasan et al. [131,132,134] with the conclusion that genistein induces pleiotropic molecular changes including down-regulation of MMP-2 and MMP-9 secretion and nuclear factor kappa B (NF-κB) DNA binding activity, inhibition of HNT invasion potential, and decreased phosphorylated AKT level, but also induced telomere shortening in vitro, altogether suggesting its potential role as a chemotherapeutic and/or chemopreventive agent for HNTs. Moreover, genistein treatment inhibited the tumorsphere formation and induced the apoptosis of nasopharyngeal cancer stem cells, enriched from CNE-2 and HONE-1 cell lines, through the suppression of sonic hedgehog (SHH) signaling activity [135]. Similarly, Hsieh et al. have demonstrated a decrease in the proliferation of HNT tumor-initiating cells, downregulation of EMT, and potentiated cell death caused by doxorubicin, cisplatin, and 5-fluorouracil (5-FU) chemotherapeutics after genistein treatment [136]. In the same study, increased ROS production induced by the upregulation of miR-34a resulted in reduced migration, invasion, self-renewal, and increased apoptosis rate of tumor-initiating cells. Furthermore, in another study on nasopharyngeal carcinoma, genistein-induced dose-dependent G2/M phase arrest in CNE-2 cell line in doses 30 to 120 µM, where SERMs (ER antagonist fulvestrant, ERα-specific agonist propyl pyrazole triol (PPT), and ERβ-specific agonist diarylprepionitrile (DPN)) did not affect genistein induced growth inhibition [137]. The growth inhibition of the SCC-25 cancer cell line via G2/M phase arrest after genistein treatment was also observed in a study conducted by Ye et al., with a significant decrease of proliferating cell nuclear antigen expression, but no difference in the number of apoptotic cells [138]. Interestingly, in vivo genistein treatment also delayed the clinicopathological change of 7,12-dimethylbenz[a]anthracene (DMBA)-induced carcinogenesis of OSCC in Syrian hamsters [139]. Moreover, Myoung et al. have demonstrated a down-regulation in VEGF mRNA expression after the genistein treatment (27.3 μg/mL) in HSC-3 cells, reduced tumor invasion through the artificial basement membrane and gelatinolytic activity compared to the control group, while in HSC-3-bearing mice treated with 0.5 mg/kg genistein, lower CD31 immunoreactivity was determined, with no difference in the tumor growth and metastatic behaviour [140]. On the other hand, in the Tu 212 laryngeal cancer cell line, genistein treatment elevates the miR-1469 expression through the activation of p53 and MCL1 inhibition at the dose of 100 μM [141]. To sum up, it can be concluded that genistein acts as an antineoplastic agent with an effect on cell cycle, apoptosis, invasion, metastasis, and angiogenesis.

Another phytoestrogen with reported potential antioxidative, anti-inflammatory, and anticancer effects in various studies is flavonoid apigenin. Several studies have demonstrated the inhibitory effect of apigenin on GLUT-1 mRNA and protein expression in different types of cancer cells, including HNTs, resulting in PI3K/AKT pathway downregulation [142,143,144,145]. Furthermore, xenograft growth inhibition and enhanced xenograft radiosensitivity as the result of suppressing the expression of GLUT-1 via the PI3K/ AKT pathway were also observed in vivo in a nude mouse model of laryngeal carcinoma [145]. Likewise, Chan et al. have suggested that apigenin may be a good therapeutic agent against HNSCC cells, since it has a similar effect as genistein: it inhibits growth, induces G2/M phase cell cycle arrest, and through the upregulation of TNF-R and TRAIL-R pathways, induces the apoptosis of the SCC-25 cell line [146]. The same effect was observed in a study published by Masuelli et al. where the survival inhibition and apoptosis induction of CAL-27, SCC-15 and FaDu HNSCC cell lines were in a dose-dependent correlation with apigenin treatment. Moreover, they have demonstrated a reduction in ligand-induced phosphorylation of EGFR and ErbB2 after the apigenin treatment, which plays a critical role in HNSCC development and progression [147].

Formononetin is an active isoflavone compound of *Astragalus membranaceus* that increased cell death by activating the caspase cascade at an IC50 value of 50 µM in the FaDu cell line, but had no effect on the viability of normal mouse fibroblasts. Furthermore, dose-dependent suppression of the mitogen-activated protein kinases ERK1/2, p38, and NF-κB phosphorylation in the same cell line were demonstrated, as well as delayed tumor growth in a FaDu cell xenograft mouse model after the oral administration of formononetin, promising that formononetin is a potential chemotherapeutic for the treatment of HNSCC [148]. Similar results have been provided in a study by Qi et al. in CNE-1 and CNE-2 nasopharyngeal carcinoma cell lines, where formononetin treatment inhibited their proliferation and induced apoptosis of CNE-1 cells, together with a decreased AKT phosphorylation, enhanced phosphorylation of c-Jun N-terminal kinase/stress-activated protein kinase (JNK/SAPK) and p38 MAPK, as well as upregulated the pro-apoptotic factors BCL2-associated X protein (BAX) and caspase-3, while downregulating the anti-apoptotic B-cell CLL/lymphoma 2 (BCL-2) [149]. In the same study, formononetin was also found to slow down the in vivo growth rate of ectopically implanted CNE-1 tumors. The same results related to the proliferation and apoptosis of the CNE-2 cell line were obtained in another study, together with a decreased wound healing process and migratory capability of the CNE-2 cells [150]. On the other hand, a contradictory result was given in a study by Guo et al. where formononetin has an inhibitory effect on apoptosis of the CNE-2 cell line by upregulating BCL-2 and p-ERK1/2 protein levels [151]. Therefore, since formononetin shows promise as an HNSCC chemotherapeutic agent, inducing apoptosis in multiple cell lines via diverse molecular pathways, conflicting results require further investigation.

Resveratrol is another well-studied phytoestrogen with a potential role in cancer chemoprevention. Interestingly, resveratrol was shown to suppress the viability and induce DNA damage in FaDu and CAL-27 cell lines in doses 5–50 µmol/L, while the Detroit562 cell line was resistant to it, even at higher dosages. Moreover, an S-phase cell cycle arrest and apoptotic cell death, together with activation of BRCA1 and γ-H2AX foci was demonstrated in FaDu and CAL-27 cells [152]. More importantly, resveratrol 24 h treatment in a non-toxic dose range (25 to 75 μM) suppressed the migration and invasion potential of the cisplatin-resistant (CAR) human OSCC cell line CAL-27 in a dose-dependent manner. Furthermore, in a 50 μM dose, resveratrol significantly decreases the expression of p-ERK and p-p38, as well as MMP-2 and MMP-9 resulting in decreased overall metastatic potential of the CAR cell line [153]. Aside from the synergistic effect and enhancing the cytotoxic effects of cisplatin, resveratrol also acts synergistically with etoposide on the induction of apoptosis and necrosis in SCC-25, CAL-27, and FaDu cell lines [154]. Resveratrol treatment in HNSCC offers many health benefits beyond just inhibition of cell growth and induction of cell death. It blocks carcinogen formation, activates antioxidant enzymes, prevents inflammation, alters DNA repair pathways, reduces resistance to chemotherapy drugs, etc. [155].

Apart from the well-documented function of previously mentioned phytoestrogens, only a few studies available explore the impact of other substances such as biochanin A, inositol-6 phosphate, apigenin, kaempferol, calycosin and other substances in the treatment of HNSCC. Treatment with isoflavone biochanin A caused dose- and time-dependent cell death of FaDu cells, together with increased activation of extrinsic (FASL and caspase-8) and intrinsic apoptotic factors (BAD and caspase-9), as well as decreased expression of intrinsic anti-apoptotic factors (BCL-2 and BCL-XL) [156]. Like the resveratrol treatment, biochanin A also inhibits wound healing potential through the suppression of MMP-2 and MMP-9, via the downregulation of p38, MAPK, NF-κB, and AKT signaling pathways [156]. On the other hand, inositol-6 phosphate showed no change in HEp-2 cells 24 and 48 h after the treatment, but 1mM dose 72 h after the treatment showed a decrease in cell number without initiating apoptosis [157]. A similar effect of reduced cell survival rate and apoptosis induction was also observed after the treatment with apigenin, kaempferol, and calycosin [158,159,160]. Table 4 summarizes the current knowledge about phytoestrogens as potential therapies for head and neck tumors.

## 8. The Role of Estrogen Signaling in Modulation of Tumor Immune Microenvironment and Microbiome Composition and thus Associated Efficacy of Immunotherapy

In order to determine their association with clinical characteristics, i.e., their role in HNSCC, the majority of studies presented in this review were based on measuring the mRNA or protein levels of nERs and mERs in tumor samples from patients. Therefore, this was primarily a tumor cell intrinsic insight into the role of ERs in HNSCC, which could be insufficient to fully characterize their activity and to identify HNSCC patients who could benefit from therapy based on antiestrogens and phytoestrogens, since the activity of these receptors is much more complex and critically depends on hormone availability, post-translational modifications, cellular localization, and protein–protein interactions [161].

Furthermore, since tumor cells do not act alone, it is also important to summarize the role of tumor microenvironment (TME) in HNSCC, with special emphasis on how E2 signaling regulates tumor immune microenvironment (TIME) and modulates this associated efficacy of immunotherapy. Extracellular matrix components and immunosuppressive cells such as cancer-associated fibroblasts (CAFs), myeloid-derived suppressor cells (MDSCs), regulatory T-cells (Tregs), and tumor-associated macrophages (TAMs) all play a crucial role in oral cancer progression, influencing tumor invasion, metastasis, drug resistance and patient prognosis [162].

TIME can either promote or inhibit the immune response against the tumor cells. For example, a TIME with active immune cells such as T lymphocytes and natural killer (NK) cells is typically more responsive to immunotherapy. On the other hand, a TIME that is characterized by immune suppression, e.g., by producing immunosuppressive cytokines, Tregs and MDSCs, or by expressing checkpoint molecules like programmed death-ligand 1 (PD-L1), can compromise the efficacy of immunotherapy [163]. Therefore, an improvement in immunotherapy outcomes requires modulation of TIME to promote immune activation, which can be performed with either cytokine therapies or immune checkpoint inhibitors (ICIs) [164].

To date, three targeted immunotherapies have been fully approved by the United States Food and Drug Administration (FDA) for the treatment of HNSCC: cetuximab, an anti-EGFR monoclonal antibody; pembrolizumab; and nivolumab, which are ICIs [165]. Through the activation of antibody-dependent cell-mediated cytotoxicity (ADCC) via NK cells and monocytes, cetuximab mediates an oncogenic signal that blocks and kills tumor cells [166]. For ten years, it was the first line of treatment for HNSCC, combined with other platinum-based chemotherapy like cisplatin and 5-fluorouracil (5-FU). Conversely, the latter two monoclonal antibodies are used to treat recurrent or metastatic (R/M) HNSCC by blocking the interaction between PD-L1 and programmed cell death protein 1 (PD-1) receptor [167]. Pembrolizumab has been approved as a first-line treatment for PD-L1-positive R/M HNSCC and in combination with chemotherapy for patients with any PD-L1 status since the results of the KEYNOTE-048 trial came out in 2019 [168]. However, although HNSCC is considered as one of the tumors with the highest frequency of PD-L1 positivity (57–82% of cases), less than 20% of HNSCC patients show an objective response to ICI treatment within the FDA settings, while most patients exhibit primary resistance [169].

So, what is the role of E2 signaling in the efficacy of immunotherapy? It is well known that male and female immune systems are different, and these differences are influenced by a variety of factors, such as genetic mediators (sex chromosomes X and Y), hormonal mediators (all three major classes of sex steroid hormones), behavioral mediators (drinking and smoking), as well as age [170]. That said, it is evident that E2 signaling could have a significant impact on TIME, since nERs expression has been observed on all aforementioned stromal and immunological components of TIME [171]. For example, ERα expression was observed in both breast and prostate CAFs, with an opposite impact on tumor invasion and macrophage infiltration—harmful in breast and positive in prostate cancer [172,173]. Furthermore, ERα expression was also observed in human ovarian cancer MDSCs, and E2-treated non-ovariectomized mice with ovarian cancer showed hastened tumor progression, with decreased levels of helper and cytotoxic T-cells, and increased concentrations of granulocytic MDSCs [174]. Likewise, cervical cancer patients who were pregnant and had high E2 levels showed increased expansion of MDSCs and shorter progression-free survival [175]. TAMs in the TME are more often M2 macrophages which secrete interleukins IL-4, IL-5, IL-6, and IL-10, known to promote tumor cell growth and immune evasion [176,177], and the highest concentration of TAMs in TIME of high-grade serous ovarian carcinomas was found in ERα-positive tumors [178]. E2 can cause immunosuppression through inhibition of cytotoxic T-cells- and NK cell-mediated tumor cell elimination [179], while E2-treated ERα-expressing CD4+CD25- T-cells regain CD25 expression, so such transformed CD4+CD25+ T-cells then manifest an immunosuppressive Treg phenotype [180].

Taken all together, all those point to the pro-tumorigenic role of E2 signaling also through its modulation of TIME, i.e., through TME immunosuppression and tumor immune evasion. Therefore, combination therapies including antiestrogen and ICI have already shown to be valuable strategies to increase the response rate of immunotherapy in nER-positive tumors, like breast cancer [181], in which FULV has shown to be the most effective [182]. By looking at the bigger picture, i.e., sex-determined differences in immune system functioning and response, patient biological sex has become a recognized predictive biomarker of response to ICIs [183]. At least in some types of cancers, like non-small-cell lung carcinoma (NSCLC) [183], since a large meta-analysis, which included results from clinical trials in advanced NSCLC, small-cell lung carcinoma (SCLC), urothelial carcinoma, HNSCC, melanoma, mesothelioma, clear cell renal cell carcinoma (ccRCC), and gastric/gastroesophageal carcinoma, has shown no study-level difference in overall survival between men and women who received immunotherapy [184]. 

Nevertheless, due to the evidently increased expression of nERs in different types of HNSCC (Table 2), sex-related differences in response to ICIs would be expected for this type of cancer. So, for example, a study which has been included in the aforementioned meta-analysis, presented survival rates stratified by patient sex, and significantly shorter OS has been observed only in males treated with ICI compared to standard chemotherapy (hazard ratio 0.65, 95% confidence interval 0.48–0.88) [185]. However, unfortunately, patient biological sex is generally very rarely reported in clinical studies conducted in HNSCC [186,187].

Therefore, it seems that the best predictive markers of response to ICIs still remain PD-L1 expression and tumor mutational burden (TMB), both of which often show higher values in males [188]. However, in HNSCC, several studies have shown either statistically insignificant differences in TMB and PD-L1 expression between men and women [169,189], while few studies even showed an increased PD-L1 expression in women [190,191].

One emerging biomarker for predicting response to ICIs, on which E2 signaling could also have impact, is the human microbiome, the entirety of microorganisms (bacteria, fungi, viruses, archaea, and protists) that colonize different parts of the human organism, or so-called body niches, like eye, ear, oral cavity, nasopharyngeal tract, gut, vagina, and skin [192]. Healthy organisms live in homeostasis (symbiosis) with beneficial bacteria; however, disease may be developed if this homeostasis is disrupted, and then some pathogenic bacteria could become more prevalent (so-called dysbiosis) [193].

As a logical niche, the role of the oral microbiome, or more precisely oral bacteriome, in oral cancer development and progression has been extensively studied in the last couple of years [194]. The five most representative oral bacterium species found in OSCC patients are *Fusobacterium periodonticum*, *Parvimonas micra*, *Streptococcus constellatus*, *Haemophilus influenza*, and *Filifactor alocis* [195]. In addition, periodontitis-related bacteria, *like Porphyromonas gingivalis*, *Fusobacterium nucleatum*, and *Prevotella intermedia*, which are associated with tooth loss and poor oral health, are also considered as a risk factor for OSCC development [196]. Molecular mechanisms by which oral microorganisms cause cancer could be divided into four groups: (I) production of carcinogenic substances (e.g., *Porphyromonas gingivalis*), (II) regulation of inflammatory and immune responses (e.g., *Fusobacterium nucleatum*), (III) promotion of cell proliferation and anti-apoptotic activity (e.g., *Streptococcus aureus*), and (IV) contribution to cellular invasiveness (e.g., *Bacteroides fragilis*) [197]. Interestingly, all those mechanisms could be controlled by E2 signaling, as we have explained previously. However, although the relationship between E2 and the first group of mechanisms could seem a little less obvious, it is well known that microbiomes could metabolize endogenous and exogenous estrogens (phytoestrogens) and thus alter the circulating estrogen levels or create estrogen metabolites, which could have an impact on hormone-dependent cancer development [198] and treatment [199]. Therefore, it is evident that the impact of E2 on the oral microbiome could cause tremendous effects on the host organism, such as changes in the immune response of the host [200].

Therefore, besides being implicated in cancer development, the great interest of both the scientific and medical communities was sparked by the discovery that bacteria composition can modulate the response to immunotherapy. One such discovery was a positive correlation between response to ICIs and the relative abundance of *Akkermansia muciniphila*, and that fecal microbiota transplantation (FMT) from cancer patients who responded to ICIs to antibiotic-treated or germ-free mice improved the antitumor effects of PD-1 blockage [201]. Since E2 can evidently modulate microbiome composition, e.g., higher E2 levels were found to be associated with a higher abundance of *Bacteroidetes* and a lower of *Firmicutes* [202], it is also known that E2 can thus modulate microbiome-related responses to ICIs. For instance, the antitumor effects of inhibition of another immune checkpoint, cytotoxic T-lymphocyte-associated protein 4 (CTLA-4), depend on the presence of specific *Bacteroides* species [203]. All these point to a tightly interwoven network of relationships between E2 signaling, TIME, and the microbiome, relationships that can have a significant impact on oral cancer development and treatment.

## 9. Conclusions and Future Perspectives

From epidemiologic studies, it is definitively evident that estrogen exposure provides a protective effect for developing HNTs in women, while menopause leads to the cessation of this effect. However, in vitro studies showed that the beneficial effect of estrogen on reducing the migration ability of tumor cells is still controversial. The full effect of estrogen is the result of a complex interplay of nuclear and membrane estrogen receptor signaling pathways. As expected, so far, primarily nuclear ERs have been studied in the context of HNTs, but those studies also show contradictory results, since different cases of the same tumor type can show significant variability in the expression of a particular form of nER or association of this expression with a survival rate. Nevertheless, the beneficial role of antiestrogens in the treatment of HNSCCs is plausible.

Since the amount of knowledge about mERs is limited, further research is needed to clarify their role and mechanisms of action in the process of carcinogenesis, separately and in integration with classical genomic estrogen signaling via nERs. One reason for the limited knowledge about mERs is that most of them are still just putative, so they lack even basic experimental tools, like antibodies, especially if they are isoforms of full-length nERs. Nevertheless, the present level of evidence suggests that the role of estrogen and ERs in HNTs is not negligible, which encourages further studying. However, to fully understand the role of ERs in HNTs, as well as the applicability of antiestrogens, alone or in combination with, e.g., immunotherapy, those receptors must be studied in the context of their complex interplay with tumor surroundings, such as the immune microenvironment or microbiome.

## Figures and Tables

**Figure 1 cancers-16-01575-f001:**
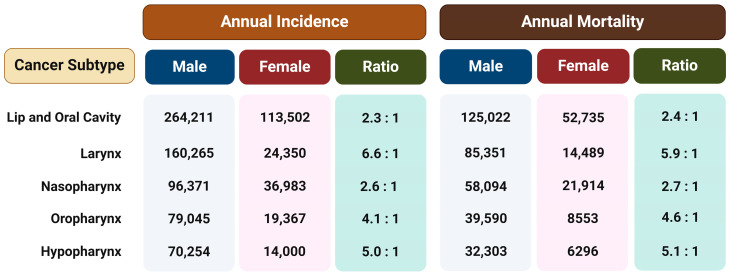
The annual incidence and mortality rates for different subtypes of head and neck tumors in both males and females and the ratio between the sexes. The data are from Global Cancer Statistics 2020 [18]. Created with https://BioRender.com (accessed on 5 January 2024).

**Figure 2 cancers-16-01575-f002:**
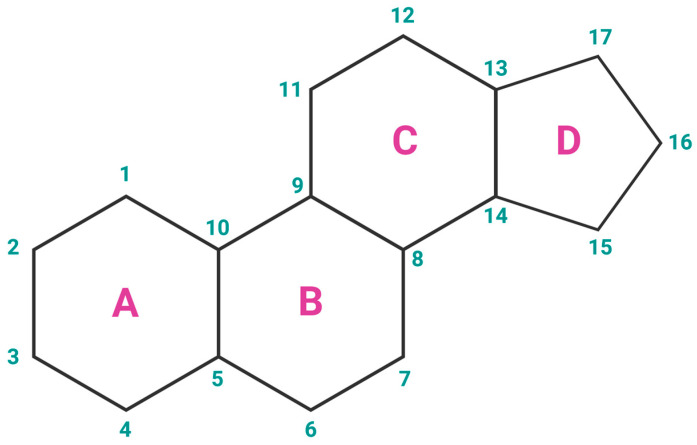
(**a**) Chemical structure of a gonane unit, the basic steroid nucleus. Steroids share the same skeleton composed of 17 carbon atoms connected in four fused rings (three cyclohexanes and one cyclopentane). The steroid ring system is labeled using IUPAC-recommended ring lettering and atom numbering. (**b**) Estrogens contain 18 carbon atoms and belong to the group of C18 steroids. They contain one phenolic hydroxyl group and one ketone group (estrone (E1)) or one (estradiol (E2)), two (estriol (E3)), or three (estetrol (E4)) hydroxyl groups on the cyclopentane ring. Created with https://BioRender.com (accessed on 5 January 2024).

**Figure 3 cancers-16-01575-f003:**
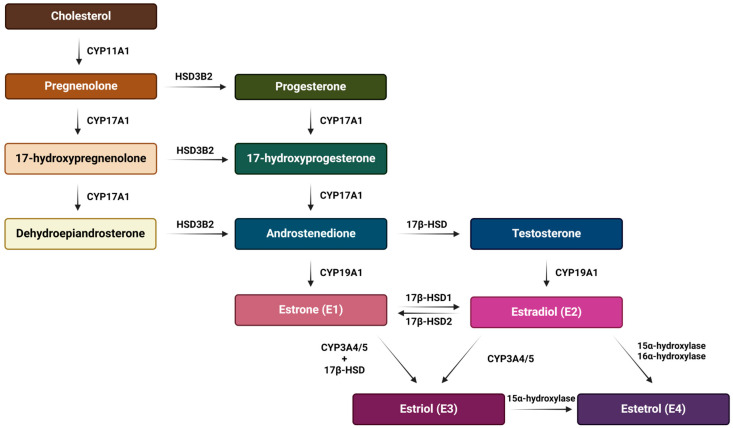
Schematic representation of the estrogen biosynthesis pathway from cholesterol to estetrol with the enzymes involved. The main synthesis pathway of the most abundant estrogens (estradiol, estrone and estriol) is their production from cholesterol and through the aromatization of androgens. On the other hand, estetrol is synthesized only during pregnancy from estradiol and estriol by fetal liver enzymes 15α and 16α-hydroxylase and it reaches the maternal circulation through the placenta. Created with https://BioRender.com (accessed on 5 January 2024).

**Figure 4 cancers-16-01575-f004:**
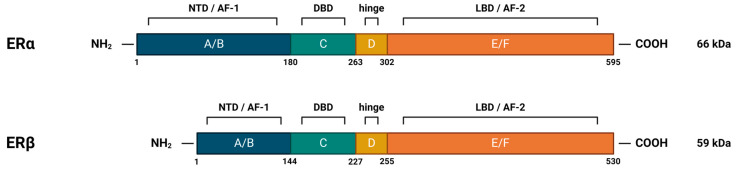
Structural and functional domains of nuclear estrogen receptors. ERα and ERβ both contain six functional domains, marked with letters A-F. The N-terminal domain (NTD), which contains domains A and B, is also the transactivation domain. C denotes the highly conserved DNA-binding domain (DBD), whose amino acid sequences in ERα and ERβ show as much as 96% identity. It is rich in cysteines and builds two functionally different zinc fingers, which enable binding to the DNA helix through specific interactions with canonical DNA sequences, called estrogen response elements (EREs), and non-specific interactions with the DNA backbone. It also contains a subdomain that allows dimerization of the receptor. D indicates the flexible hinge domain (HD), which contains the nuclear localization signal. The C-terminal domain, marked E/F, contains a ligand-binding domain (LBD) with a hydrophobic cavity (~75% hydrophobic side chains) in which the ligand can be placed and the receptor can be allosterically activated. The ligand binding site contains key polar side chains of arginine and glutamic acid, which form hydrogen bonds with the hydroxyl group on the A-ring of the ligand and are highly conserved in nuclear steroid receptors. In the case of the estrogen receptor, an additional hydrogen bond with the D-ring of estrogen is achieved via the histidine side chain. NTD and LBD also contain activation function regions (AF-1 and AF-2, respectively), by means of which the receptor, after binding to DNA, recruits coregulatory proteins. Created with https://BioRender.com (accessed on 5 January 2024).

**Figure 5 cancers-16-01575-f005:**
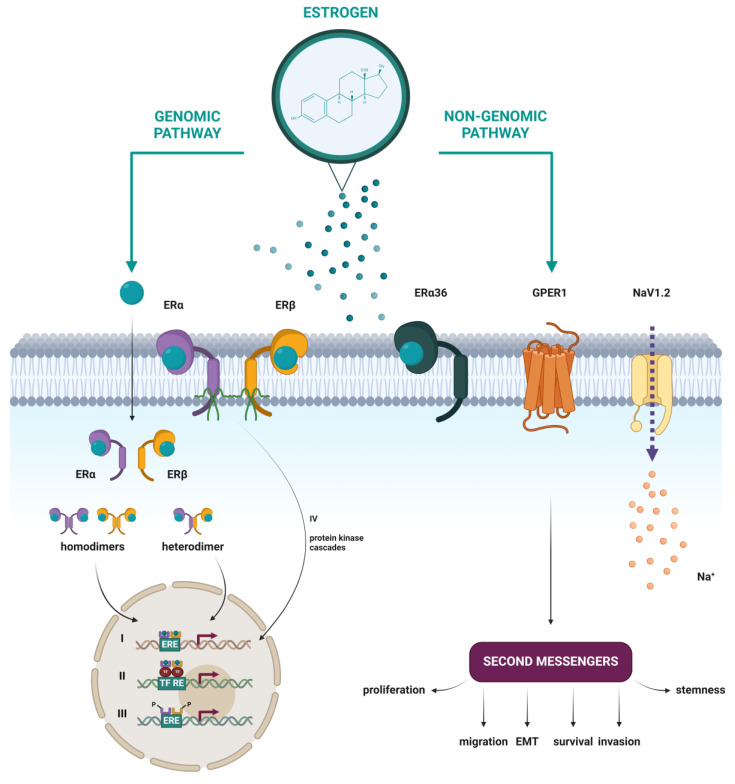
Schematic representation of nuclear and membrane estrogen receptors and different mechanisms of estrogen action through genomic and non-genomic pathways. (I) Direct genomic signaling in which the E2-ER complex directly binds to the ERE and activates gene expression. (II) Indirect genomic signaling, where the E2-ER complex regulates expression by protein–protein interactions with other transcription factors at their respective response elements. (III) Ligand-independent signaling, in which activation occurs by phosphorylation of ERs or their co-regulators. (IV) Non-genomic signaling, which includes the activation of various protein kinase cascades in response to E2 binding to membrane forms of ERα and ERβ or to the membrane estrogen receptors ERα36, GPER1, and NaV1.2. Created with https://BioRender.com (accessed on 5 January 2024).

**Table 1 cancers-16-01575-t001:** Reference ranges for serum and urine estradiol (data from [33]).

		Serum (pg/mL)	Urine (µg/24 h)
Child (<10 years old)		<15	0−6
Adult male		10−50	0−6
Adult female	Follicular phase	20−350	0−13
	Midcycle peak	150−750	4−14
	Luteal phase	30−450	4−10
	Post-menopause	≤20	0−4

**Table 2 cancers-16-01575-t002:** Currently known significance of estrogen receptors in head and neck tumors.

Type of Estrogen Receptor	Receptor	Receptor Function	Significance in HNTs	Reference
Nuclear	ERα, ERβ	Transcription factor	Increased ERβ expression in HNSCC compared to normal tissue, no difference in expression between males and females, neither in tumors nor in normal tissue	[80]
			Contradictory findings: Predominant ERβ expression in most OSCC studies; Predominant ERα expression in oral cavity and laryngeal/hypopharyngeal cancers	[81,82,83,84,85]
			Frequent co-expression (~40% of cases) of ER and progesterone receptors independent of primary tumor site	[85]
			Higher level of both *ESR1* and ERα in laryngeal cancer	[86]
			Mutually opposite effect in same type of HNT: ERα induces, ERβ inhibits growth of papillary thyroid cancer	[87,88]
			ERβ expression in OPSCC associated with higher survival rates compared to ERβ-negative OPSCC	[89]
			Higher ERα expression associated with improved survival rates in OPSCC patients receiving primary chemoradiation; ERα as a biomarker for better overall survival in patients with HPV+ OPSCC	[90,91]
			Significant influence of ERα expression on decrease in overall and relapse-free survival only in male OSCC cohort, in comparison to ERα-negative patients	[92]
			HPV positivity and smaller HNSCC tumor size (≤T2) were independently associated with ERα positivity	[93]
			Elevated expression of FAK is associated with increased ERα phosphorylation, transcription, and cell growth of OSCC cells	[95]
			Increased ERβ expression or ERβ agonist treatment inhibited SCC cells proliferation and promoted NOTCH1 expression in vitro and in mouse xenotransplants	[96]
			ERβ expressed in majority of laryngeal carcinomas (83%); expression is positively correlated with maintenance of E-cadherin and ß-catenin at cell junctions of tumor cells plasma membranes, and negatively correlated with increased TNM stage, nuclear translocation of β-catenin, and loss of E-cadherin	[97]
Membrane	ERα36	Membrane-initiated estrogen signaling	Increases PKC activity in laryngeal cancer cells, increasing proliferation and survival, and enhancing expression of metastatic and angiogenic factors in response to E2; ERα36 expression positively correlated with VEGF in laryngeal tumor samples and with metastasis to regional lymph nodes	[50]
			Inverse correlation between ERα66 and ERα36 expression and clinical cancer stage: ERα66 was decreased with ascending tumor aggressiveness, while ERα36 expression was increased	[104,105]
	GPER1	G protein-coupled receptor	Upregulates IL-6, promoting proliferation and migration of LSCC cells in response to estrogen mimetic bisphenol A	[98]
			Elevated expression in HNSCC compared to normal tissue; Lower expression was associated with poor prognosis	[99]
			GPER1 antagonist G15 shows antitumor effect in OSCC cell lines (SCC-4, SCC-9, HSC-3): induces dose-dependent cytotoxicity, G2/M cycle arrest, and apoptosis; downregulates expression of AKT, cell cycle-related proteins, and mitogen-activated protein kinases; induces formation of autophagosomes	[100]
	NaV1.2	Sodium ion channel	HPV viral integration into *SCN2A* genomic region observed in oral and oropharyngeal cancers: fusion of HPV L2 gene into *SCN2A* intron 16 results in gene disruption and homozygous loss of *SCN2A*	[102]
			Upregulated *SCN2A* expression in smoking HNSCC patients compared to never-smokers	[103]

FAK—focal adhesion kinase; HNSCC—head and neck squamous cell carcinoma; HNT—head and neck tumor; HPV+—human papillomavirus-positive; IL-6—interleukin-6; LSCC—laryngeal squamous cell carcinoma; OPSCC—oropharyngeal squamous cell carcinoma; OSCC—oral squamous cell carcinoma; PKC—protein kinase C; SCC—squamous cell carcinoma; VEGF—vascular endothelial growth factor.

**Table 3 cancers-16-01575-t003:** Antiestrogens as potential therapies for head and neck tumors.

Antiestrogen	Effect on HNTs	Dose	Cell Lines	Reference
Tamoxifen	Inhibits growth of laryngeal cancer cells in vitro and in vivo	3–8 µM	UM-SCC-5, UM-SCC-11B	[75]
	Inhibits growth of ER-negative HNT cells	5–10 µM	UM-SCC-11B, UM-SCC-14C, UM-SCC-22B	[109]
	Insensitivity to TAM treatment observed in three ER-negative HNT cells, while it had an inhibitory effect on three ER-positive HNT cells	5 µM	UM-SCC-5, UM-SCC-9, UM-SCC-12; UM-SCC-1, UM-SCC-3, UM-SCC-14B	[110]
	Proliferation inhibition of SCC cells by inhibiting G1/S phase progression; this inhibition correlated with the upregulation of p27 and downregulation of cyclin E and CDK6	100 nM	SCC-4, SCC-9, SCC-25	[111]
	Inhibits invasion of SCC cells and induces anoikis as a direct result of adhesion inhibition and disruption of survival signals, due to the reduction in phosphorylation of FAK, ERK, and MAPK	3–30 µM	SCCTF, SCCKN, SAS, Ca9-22	[82]
	TAM in combination with cisplatin enhanced cytotoxic and apoptotic effect on OSCC cells, possibly through inhibition of PKC activity and upregulation of the TGFB1	5 µM TAM, 5 µg/mL cisplatin	A-253, HSC-3, KB	[108]
	Induced G1 cell cycle arrest independently of p53 status and increased level of hypophosphorylated active RB in OSCC; TAM combined with cisplatin induced apoptosis more effectively and resulted in increased secretion of TGFB1	1 µM TAM, 5 µg/mL cis-platin	HN-6, HN-5	[112]
	Delayed development of cisplatin resistance in HNT cells	3.5 µM TAM, 6.5 µM cisplatin	UM-SCC-10B	[113]
	Induced growth inhibition and increased the OSCC cells aggregation ability	5 µM	UM-SCC-14A, UM-SCC-14B, UM-SCC-14C	[114]
	Significantly sensitized HNSCC cells to fractionated irradiation (IR)	1 µM	FaDu	[121]
	Reduced β1 integrin transcription and α3 integrin cell surface expression and inhibited the growth of OSCC cells	1, 5 µM	UM-SCC-14A, UM-SCC-14B, UM-SCC-14C	[122]
Fulvestrant	Significantly sensitized HNSCC cells to fractionated irradiation (IR)	10 nM	FaDu	[121]
	Reduced laminin-1 adhesion and inhibited growth of OSCC cells	1, 5 µM	UM-SCC-14A, UM-SCC-14B, UM-SCC-14C	[122]
	Restored estrogen-mediated decrease of apoptosis in pre-malignant oral leukoplakia cells	1 µM	MSK-Leuk1	[80]
Centchroman	Antiproliferative effect in HNSCC cells; induces apoptosis and inhibits AKT/mTOR and STAT3 signaling; inhibits colony formation of HNSCC cells and alters proteins associated with DNA damage and cell cycle regulation	2.5, 5, 10 µM	FaDu, CAL-27, SCC-9, SCC-25	[125]

ERK—extracellular signal-related kinase; FAK—focal adhesion kinase; HNSCC—head and neck squamous cell carcinoma; HNT—head and neck tumor; MAPK—mitogen-activated protein kinase; OSCC—oral squamous cell carcinoma; PKC—protein kinase C; RB—retinoblastoma protein; SCC—squamous cell carcinoma; TAM—tamoxifen; TGFB1—transforming growth factor beta-1.

**Table 4 cancers-16-01575-t004:** Phytoestrogens in HNT cancer prevention/treatment/pathogenesis.

Phytoestrogen	Effect on HNTs	Dose	Cell Line or Organism	Reference
Genistein	Induced time-dependent and irreversible proliferation inhibition, S/G2-M phase cell cycle arrest and apoptosis	5–50 µM	HN4	[131]
	Down-regulation of CDK1 and cyclin B1, up-regulation of CDK inhibitor p21WAF1 and apoptosis regulator BAX	25 µM	HN4	[132]
	Treatment with genistein nanoparticles selectively induced apoptosis by increasing ROS production, leading to translocation of BAX proteins to mitochondria and caspase 3 activation	40 µM	JHU-011	[133]
	Down-regulation of MMP-2 and MMP-9 secretion and NF-κB DNA binding activity, inhibition of HNT invasion potential, decreased level of phosphorylated AKT, induced telomere shortening	5–50 µM	HN4	[134]
	Inhibited tumorsphere formation and induced apoptosis of nasopharyngeal cancer stem cells, through the suppression of SHH signaling	100 μM	CNE-2, HONE-1	[135]
	Decreased HNT TICs proliferation; downregulation of EMT; potentiated cell death caused by doxorubicin, cisplatin, 5-FU chemotherapeutics; increased ROS production induced by miR-34a, which resulted in reduced migration, invasion, self-renewal, and increased apoptosis rate	20 μM	HNT-TICs	[136]
	Induced dose-dependent G2/M phase arrest, where SERMs (FULV, propyl pyrazole triol, diarylprepionitrile) did not affect genistein-induced growth inhibition	30–120 µM	CNE-2	[137]
	Growth inhibition via G2/M phase arrest, decreased proliferating cell nuclear antigen expression, no difference in number of apoptotic cells	50–200 µM	SCC-25	[138]
	Delayed the clinicopathological change of DMBA-induced carcinogenesis of OSCC in vivo	20 mg/kg	Syrian hamster	[139]
	Down-regulation in *VEGF* mRNA expression, reduced tumor invasion through artificial basement membrane and gelatinolytic activity in vitro; Lower CD31 immunoreactivity in vivo, no difference in tumor growth and metastatic behaviors	27.3 μg/mL; 0.5 mg/kg	HSC-3; HSC-3 cell xenograft mouse model	[140]
	Elevated miR-1469 expression through p53 activation MCL1 inhibition	100 μM	Tu 212	[141]
Apigenin	Dose-dependent inhibitory effect on *GLUT-1* mRNA and protein expression, resulting in the PI3K/AKT pathway downregulation in cisplatin-treated HNT cells	40–160 µM	HEp-2	[144]
	Xenograft growth inhibition and enhanced xenograft radiosensitivity as the result of suppressing GLUT-1 expression via the PI3K/AKT pathway in vivo	Intraperitoneal injection with 50 or 100 µg	HEp-2 cell xenograft mouse model	[145]
	Inhibits growth, induces G2/M phase cell cycle arrest, induces apoptosis through upregulation of TNF-R and TRAIL-R pathways	10, 20 µM	SCC-25	[146]
	Dose-dependent survival inhibition and apoptosis induction; reduction in ligand-induced phosphorylation of EGFR and ErbB2	6–100 µM; 50 µM	CAL-27, SCC-15, FaDu	[147]
Formononetin	Increased cell death by activation of caspase cascade; dose-dependent suppression of the mitogen-activated protein kinases ERK1/2 and p38, and NF-κB phosphorylation in vitro; delayed tumor growth in vivo after oral administration	5–50 µM; 10 mg/kg	FaDu; FaDu cell xenograft mouse model	[148]
	Inhibited proliferation and induced apoptosis, decreased AKT phosphorylation, enhanced phosphorylation of JNK/SAPK and p38 MAPK, upregulated pro-apoptotic factors BAXand caspase 3, and downregulated the anti-apoptotic BCL-2; slowed down tumor growth rate in vivo	5–40 µM; intraperitoneal injection with 10 or 20 mg/kg	CNE-1; CNE-1 cell xenograft mouse model	[149]
	Inhibited proliferation and induced apoptosis, decreased wound healing process and migratory capability	10, 20, 40 μM	CNE-2	[150]
	Inhibitory effect on apoptosis by upregulating BCL-2 and p-ERK1/2	0.1–1 µM	CNE-2	[151]
Resveratrol	Suppressed viability and induced DNA damage, induced S-phase arrest and apoptosis, together with activation of BRCA1 and γ-H2AX foci	5–50 µM	FaDu, CAL-27	[152]
	Suppressed migration and invasion potential of cisplatin-resistant human OSCC cells in a dose-dependent manner; decreased expression of p-ERK, p-p38, MMP-2, and MMP-9, resulting in decreased overall metastatic potential	25, 50, 75 µM; 50 μM	cisplatin-resistant CAL-27	[153]
	Synergistic effect with etoposide on the induction of apoptosis and necrosis	20–240 μM, etoposide:resveratrol = 1:4	CAL-27, SCC-25, FaDu	[154]
Biochanin A	Induced dose- and time-dependent cell death; increased activation of extrinsic (FASL and caspase-8) and intrinsic apoptotic factors (BAD and caspase-9), decreased expression of intrinsic anti-apoptotic factors (BCL-2 and BCL-XL); inhibited wound healing potential through MMP-2 and MMP-9 suppression, via downregulation of p38, MAPK, NF-κB, and AKT signaling pathways	25, 50 µM	FaDu	[156]
Inositol-6 phosphate	Decrease in cell number without initiating apoptosis	1 mM	HEp-2	[157]
Kaempferol	Dose-dependent decrease in cell viability	0.1–100 µM	FaDu	[160]
Calycosin	Dose-dependent reduction of cell survival rate and increased apoptosis in vitro; upregulated expression of *TP53* and CASP8, and reduced MAPK14 expression in vitro and in vivo; dose-dependent reduction in tumor mass in vivo	20, 40, 80 µM; 20, 30, 60 mg/kg	CNE-1; CNE-1 cell xenograft mouse model	[159]

5-FU—5-fluorouracil; BAD—BCL-2-associated death promoter; BAX—BCL2-associated X protein; BCL-2—B-cell CLL/lymphoma 2; BCL-XL—B-cell lymphoma-extra-large; BRCA1—breast cancer gene 1; CD31—cluster of differentiation 31; CDK1—cyclin-dependent kinase 1; DMBA—7,12-dimethylbenz[a]anthracene; EGFR—epidermal growth factor receptor; EMT—epithelial–mesenchymal transition; FULV—fulvestrant; GLUT-1—glucose transporter 1; HNT—head and neck tumor; JNK/SAPK—c-Jun N-terminal kinase/stress-activated protein kinase; MAPK—mitogen-activated protein kinase; MCL1—myeloid cell leukemia-1; MMP—matrix metalloproteinase; NF-κB—nuclear factor kappa B; OSCC—oral squamous cell carcinoma; ROS—reactive oxygen species; SERMs—selective estrogen receptor modulators; TICs—tumor-initiating cells; TNF-R—tumor necrosis factor receptor; TRAIL-R—TNF-related apoptosis-inducing ligand receptor; *VEGF*—vascular endothelial growth factor.

## Data Availability

No new data were created or analyzed in this study. Data sharing is not applicable to this article.

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
