# Peer review of "The Role of Estrogen and Estrogen Receptors in Head and Neck Tumors"

_cancers, 2024, doi:10.3390/cancers16081575_

Round 1

Reviewer 1 Report

Comments and Suggestions for Authors

The current manuscript described the fundamental knowledge of estrogen and estrogen receptors, especially roles in head and neck rumors. The differences of incidence and mortality between men and women makes the authors think that estrogen and estrogen receptor might have high impacts, which is reasonable and worthy of further study the relationships and development of possible therapies.

This review paper introduced the unique features of head and neck tumors, disease differences between genders, role of estrogen and its receptors in head and neck tumors, and related research results of antiestrogens for HNTs. I believe the above chapters are valuable to readers and researchers of the related scientific directions, which also make the current manuscript valuable to be published in this journal.

However, I would like to make some suggestions for modifications:

1. The 3rd section of "functions of estrogen and estrogen receptors" is somewhat too long, although a lot of useful knowledge about estrogen were introduced, it makes the paper tedious;

2. In the 7th section, more detailed data should be provided. For examples, different EC50s of IC50s for different cell lines should be quoted in the manuscript. Two column could be added to table 2, one for drug name and another for activity data.

3. No in vivo data. If the authors could find some in vivo data for estrogen related therapy, that will be more practical.

4. In section 2, only one subtitle "2.1 risk and factors" was listed. if there were no more subtitles, "2.1" could be deleted.

Author Response

  1. The 3rd section of "functions of estrogen and estrogen receptors" is somewhat too long, although a lot of useful knowledge about estrogen were introduced, it makes the paper tedious;

As suggested, we have shortened the text in that section.

  1. In the 7th section, more detailed data should be provided. For examples, different EC50s of IC50s for different cell lines should be quoted in the manuscript. Two column could be added to table 2, one for drug name and another for activity data.

As suggested, we have added an additional column in Table 3 (former Table 2), since the drug name is already written in the first column. However, since only  two papers explicitly stated IC50/LD50 values (Refs. 114 and 126, former 109 and 121, respectively), in that new column we rather provided all doses used in a study.

  1. No in vivo data. If the authors could find some in vivo data for estrogen related therapy, that will be more practical.

It turns out that only one study (Ref. 76, former 71) conducted in vivo experiments related to  estrogen therapy in HNSCC, and that was stated in both text and now Table 3.

  1. In section 2, only one subtitle "2.1 risk and factors" was listed. if there were no more subtitles, "2.1" could be deleted.

As suggested, we have deleted the title of section 2.1.

Reviewer 2 Report

Comments and Suggestions for Authors

The review article by Kranjčević et al. provides a comprehensive overview of the current knowledge on the role of estrogen and estrogen receptors (ERs) in the development and progression of head and neck cancer, with special emphasis on membrane ERs, which are much less studied. The manuscript is well written, but the authors should add a Materials and Methods chapter with a detailed description of the screening strategy (e.g., keywords used, literature sources, how many papers in what time period) and selection criteria for prioritizing the papers discussed in this review article. Other suggestions to improve the quality and impact of this manuscript are as follows:

1. The statement "that exposure to the female sex hormone estrogen in women may provide protection against the development of these tumors in women" (lines 14-16, 127-129) is oversimplified and not supported by compelling experimental evidence. Sex differences in cancer incidence and mortality, including head and neck cancer, are more likely the result of a complex interplay between sex chromosomes, sex hormones, and other biological and lifestyle factors.

2. Most retrospective studies of tumor samples from HNT patients have evaluated the expression of nERs and mERs at the transcript or protein level to infer their association with clinical features. As discussed in this manuscript, the activity of these receptors is much more complex and critically depends on hormone availability, post-translational modifications, cellular localization, and protein-protein interactions. Therefore, the mere detection of transcript or protein levels in tumor tissue is insufficient to describe their activity and to select HNT patients who might benefit from endocrine therapy, and should be discussed as a critical issue in the manuscript.

3. In this review, the authors provide a strong tumor cell intrinsic view of the regulation and function of nERs and mERs. However, nERs and mERs are also expressed by stromal cells of the tumor microenvironment (TME), particularly immune cells, and their estrogen-dependent and -independent activity has been implicated in immune surveillance and escape in many cancers, including HNTs. The role of estrogen signaling, nERs, and mERs in the reciprocal interaction of tumor cells with their TME and their potential impact on the efficacy of immunotherapy (e.g., sex differences in response) needs to be discussed.

Comments on the Quality of English Language

The manuscript is well written, and only minor editing of the English languegae is recommended.

Author Response

The review article by Kranjčević et al. provides a comprehensive overview of the current knowledge on the role of estrogen and estrogen receptors (ERs) in the development and progression of head and neck cancer, with special emphasis on membrane ERs, which are much less studied. The manuscript is well written, but the authors should add a Materials and Methods chapter with a detailed description of the screening strategy (e.g., keywords used, literature sources, how many papers in what time period) and selection criteria for prioritizing the papers discussed in this review article. Other suggestions to improve the quality and impact of this manuscript are as follows:

Since this is not a systematic review and since, according to the reviewers’ suggestions, we now included some additional, more broader topics than first planned, we think that adding a Materials and Methods chapter with all used search terms should not bring an added value to this literature review manuscript. Especially since all usable/findable papers were included and could be found in the reference list.

  1. The statement "that exposure to the female sex hormone estrogen in women may provide protection against the development of these tumors in women" (lines 14-16, 127-129) is oversimplified and not supported by compelling experimental evidence. Sex differences in cancer incidence and mortality, including head and neck cancer, are more likely the result of a complex interplay between sex chromosomes, sex hormones, and other biological and lifestyle factors.

Papers with experimental evidence suggesting a protective role of estrogen in women were cited in a manuscript before, while a more detailed description of a complex interplay of sex chromosomes, sex hormones, and other biological and lifestyle factors in HNSCC incidence and mortality is added in lines 45-60.

  1. Most retrospective studies of tumor samples from HNT patients have evaluated the expression of nERs and mERs at the transcript or protein level to infer their association with clinical features. As discussed in this manuscript, the activity of these receptors is much more complex and critically depends on hormone availability, post-translational modifications, cellular localization, and protein-protein interactions. Therefore, the mere detection of transcript or protein levels in tumor tissue is insufficient to describe their activity and to select HNT patients who might benefit from endocrine therapy, and should be discussed as a critical issue in the manuscript.

Thanks for the suggestion that is a very important issue to discuss in the manuscript. By taking into consideration also your 3rd commentary and suggestion, we have added a new section 8 on the importance and significance of tumor surroundings.

  1. In this review, the authors provide a strong tumor cell intrinsic view of the regulation and function of nERs and mERs. However, nERs and mERs are also expressed by stromal cells of the tumor microenvironment (TME), particularly immune cells, and their estrogen-dependent and -independent activity has been implicated in immune surveillance and escape in many cancers, including HNTs. The role of estrogen signaling, nERs, and mERs in the reciprocal interaction of tumor cells with their TME and their potential impact on the efficacy of immunotherapy (e.g., sex differences in response) needs to be discussed.

As replied previously, we have added a new section 8 on the role of estrogen signaling in modulation of tumor immune microenvironment and thus related efficacy of immunotherapy in HNSCC.

Reviewer 3 Report

Comments and Suggestions for Authors

In this review, the authors focus on “the Role of Estrogen and Estrogen Receptors in Head and Neck

Tumors”. They discuss the link between the gender and risk of head and neck tumours (HNT), propose that such link might be due to the role of estrogen and it's receptors in HNT pathogenesis. They provide detailed information regarding all known (so far) receptors of estrogen, both membrane and nuclear, their role in cancer pathogenesis, disscuss the potential of anti-estrogen therapies.

Please find my comments and suggestions below.

The authors start the review from suggesting that gender difference in cancer rates (in case of some cancers and HNT in particular) might be explained by the role of sex hormones.

What are the other possible explanations? Please, briefly discuss and provide references.

The reference [7] states that “In general, the high male to female ratios for HPV-negative HNSCC incidence reflect the sex-specific patterns of modifiable risk behaviours”. This is contradictory to that the authors say in the text about male gender per se, independently of any other factors, is associated with higer risk of HNT.

It would be beneficial to provide a summary of the levels of estrogen (in blood/other tissues?) in men vs women, at different stages of life, perhaps in a form of table (it might also increase number of quotations of the manuscript, as such table would be of use for many readers).

Do the authors want to discuss the role of phytoestrogens (mimicing the effects of human estrogen) in cancer prevention/treatment/pathogenesis?

What about link between estrogen-microbiome-cancerogenesis, giving the important role of microbiome in HNC?

Briefly, what is the role of estrogen and ER in modulation of the immunotherapy response in HNC?

Overall, I find this review comprehensive and well-written.

Author Response

The authors start the review from suggesting that gender difference in cancer rates (in case of some cancers and HNT in particular) might be explained by the role of sex hormones.

What are the other possible explanations? Please, briefly discuss and provide references.

Another possible explanation is that sex differences in HNSCC incidence and mortality are more likely the result of a complex interplay between sex chromosomes, sex hormones, and other biological and lifestyle factors, which is explained in more detail in lines 45-60.

The reference [7] states that “In general, the high male to female ratios for HPV-negative HNSCC incidence reflect the sex-specific patterns of modifiable risk behaviours”. This is contradictory to that the authors say in the text about male gender per se, independently of any other factors, is associated with higer risk of HNT.

As mentioned above, we agree that male gender per se is not an independent risk factor in HNSCC tumorigenesis, rather the sex differences in HNSCC incidence and mortality are result of a complex interaction of many biological and lifestyle factors, including HPV infections, but sex hormones and their receptors play an important role in HNSCC tumorigenesis for sure.

It would be beneficial to provide a summary of the levels of estrogen (in blood/other tissues?) in men vs women, at different stages of life, perhaps in a form of table (it might also increase number of quotations of the manuscript, as such table would be of use for many readers).

Thank you for your suggestion. We have added new Table 1 with reference ranges for serum and urine estradiol in child, men, and women.

Do the authors want to discuss the role of phytoestrogens (mimicing the effects of human estrogen) in cancer prevention/treatment/pathogenesis?

We thank the reviewer for a suggestion to discuss the role of phytoestrogens in HNSCC. A new subchapter 7.2. about phytoestrogens was incorporated into the manuscript.

What about link between estrogen-microbiome-cancerogenesis, giving the important role of microbiome in HNC?

Briefly, what is the role of estrogen and ER in modulation of the immunotherapy response in HNC?

As suggested, we have added a new section on the role 8 of estrogen signaling in modulation of microbiome and tumor immune microenvironment and thus related efficacy of immunotherapy in HNSCC.

Overall, I find this review comprehensive and well-written.

Thanks!!

Round 2

Reviewer 2 Report

Comments and Suggestions for Authors

The authors adequately addressed most of the suggestions to improve the quality and impact of the manuscript, with the exception of adding a Materials and Methods chapter.